# Exosomes: Small Vesicles with Important Roles in the Development, Metastasis and Treatment of Breast Cancer

**DOI:** 10.3390/membranes12080775

**Published:** 2022-08-12

**Authors:** Ling’ao Meng, Kedong Song, Shenglong Li, Yue Kang

**Affiliations:** 1Department of Breast Surgery, Cancer Hospital of China Medical University, Liaoning Cancer Hospital & Institute, Shenyang 110042, China; 2State Key Laboratory of Fine Chemicals, Dalian R&D Center for Stem Cell and Tissue Engineering, Dalian University of Technology, Dalian 116024, China; 3Department of Bone and Soft Tissue Tumor Surgery, Cancer Hospital of China Medical University, Liaoning Cancer Hospital & Institute, Shenyang 110042, China

**Keywords:** exosome, breast cancer, drug resistance, targeted delivery, cancer therapeutics

## Abstract

Breast cancer (BC) has now overtaken lung cancer as the most common cancer, while no biopredictive marker isolated from biological fluids has yet emerged clinically. After traditional chemotherapy, with the huge side effects brought by drugs, patients also suffer from the double affliction of drugs to the body while fighting cancer, and they often quickly develop drug resistance after the drug, leading to a poor prognosis. And the treatment of some breast cancer subtypes, such as triple negative breast cancer (TNBC), is even more difficult. Exosomes (Exos), which are naturally occurring extracellular vesicles (EVs) with nanoscale acellular structures ranging in diameter from 40 to 160 nm, can be isolated from various biological fluids and have been widely studied because they are derived from the cell membrane, have extremely small diameter, and are widely involved in various biological activities of the body. It can be used directly or modified to make derivatives or to make some analogs for the treatment of breast cancer. This review will focus on the involvement of exosomes in breast cancer initiation, progression, invasion as well as metastasis and the therapeutic role of exosomes in breast cancer.

## 1. Introduction

Female breast cancer (BC) incidence has surpassed lung cancer as the most common cancer and the fifth leading cause of cancer death worldwide in 2020 [1]. It ranked first for incidence and mortality among women in the vast majority of countries. However, pre clinical predictive and prognostic markers for already approved use have not yet emerged in BC, and, therapeutically, chemotherapy is often ineffective due to increased drug resistance, and targeted therapies, while highly effective, are not currently available for triple negative breast cancer (TNBC). These present a formidable problem for clinicians.

Extracellular vesicles (EVs) are a heterogeneous group of cell-derived membrane structures ranging between 30–10,000 nm in diameter, and EVs can be classified according to their biosynthesis or release pathway [2,3,4,5]: Exosomes (Exos) originating in the endocytic pathway, 40–160 nm in diameter (average diameter ∼100 nm); Ectosomes, approximately 50–1000 nm in diameter, are released directly from the plasma membrane, including microparticles and microvesicles (MVs); Apoptotic bodies (APOs), generated by apoptosis, are 50 nm–2 μm in diameter; Large oncosomes (LOs), generated by their release from tumor cells, are 1–10μm in diameter; and various other subpopulations of EVs.

EVs are derived from the endosomal system or released from the plasma membrane and can generate intersections with other intracellular vesicles and organelles, thereby increasing their diversity. They can utilize differential centrifugation, ultracentrifugation, density (sucrose) centrifugation method, tangential flow filtration and variations thereon, field-flow fractionation, asymmetric flow field-flow fractionation, field-free viscoelastic flow, alternating current electrophoretics, size exclusion chromatography, ion exchange chromatography, microfiltration, fluorescenceactivated sorting, deterministic lateral displacement arrays, ultrafiltration, immunoisolation, co-precipitation method and other methods [6] have been isolated EVs from various cell conditioned medium (CM), biological fluids (including blood, urine, cerebrospinal fluid, lymph, tears, saliva and nasal secretions, ascites and semen) and tissues, but there is no general consensus on the optimal isolation method [7].

Depending on the cell of origin, EVs can contain many components of the cell, including DNA, RNA, lipids, cellular metabolites, cytosol and cell surface proteins. Studies have found that EVs are widely involved in intercellular communication and have the ability to promote tissue regeneration, thus promising as an alternative to cell therapy compared to transplanted stem cells that can evade immune responses; because of its origin from membranes, it can easily cross biological barriers, protect internal carriers from degradation, and participate in the transfer of bioactive components. These properties determine its potential as a potential therapeutic agent. EVs can be used both in their native form and as vehicles for therapeutic agents as well as vector platforms to construct. But there are still many obstacles to overcome for EVs as a therapeutic means for widespread clinical application [8]. A large number of studies have proved that EVs can promote the occurrence, development, prediction, diagnosis and treatment of diseases in many human systems: nervous system diseases [9,10,11,12,13], cardiovascular system [14,15,16,17,18,19,20,21], metabolic system [22,23], motor system [24,25,26,27,28,29], female reproductive system [30,31], and various tumors [32,33,34,35]. This review will focus on exosomes in EVs for the occurrence and progression of breast cancer, as biomarkers to predict breast cancer, and exosome-based novel breast cancer therapies.

## 2. The Biogenesis, Biomarkers and Contents of Exosomes

The term “exosome” was first named in 1987 after Johnstone et al. observed during the maturation of sheep reticulocytes cultured in vitro [36]. It is a subset of EVs originating in the endocytic pathway with a diameter of 40–160 nm and is one of the most studied EVs (Figure 1). Exosomes production involves the double invagination of the plasma membrane and the formation of multivesicular bodies (MVB) containing intraluminal vesicles (ILV). ILV is eventually secreted as exosomes, with a size range of about 40 to 160 nm in diameter, and is fused to the plasma membrane by MVB and Exos exchange and transfer biological information through autocrine, parocrine and endocrine modes. Which is derived from the cells of the cellular structure of vesicles, has a good biocompatibility, evade the phagocytosis, high stability, no immunogenicity, and can be used as a tumor biological forecast is widely used in screening for clinicians, in the process of targeted drug delivery, because its are natural nanometer carrier, with small diameter, can cross the barrier of cells, such as the blood-brain barrier (often applied to tumors of the nervous system). These excellent physical, chemical and biochemical characteristics stand out among natural nanocarriers. These cell-derived nanovesicles with intrinsic biological functions are highly competent for the establishment of the next generation of nanomedicine [37]. Their biomarkers include CD9, CD81, CD63, ceramide, flotillin, its internal Alix, ARF6, TSG101 [3].

The contents include various mRNA, microRNA (miRNA) and other non-coding RNA, DNA, lipid, cytoplasmic and membrane proteins, including receptors and major histocompatibility complex (MHC) molecules [38], etc.

## 3. Exosomes and Breast Cancer Microenvironment

### 3.1. Exosomes Promote the Occurrence, Development and Metastasis of Breast Cancer

#### 3.1.1. Exosomes Weaken Tumor Immune System and Enhance Immune Escape of Breast Cancer Cells

The role of exosomes in tumor microenvironment (TME) has been widely studied, and they have the ability to promote the occurrence, development and metastasis of tumor.

Immunosuppression is a major obstacle to effective immunotherapy for cancer. Interleukin-6 (IL-6) is an important trigger for the expansion and recruitment of myeloid-derived suppressor cells (MDSC), which is considered to be the primary coordinator of the immunosuppressive TME. Studies have shown that IL-6 has the ability to promote the development of breast cancer [39,40,41], while early-stage myeloid-derived suppressor cells (eMDSCs) have a stronger immunosuppressive ability [42].

MiR-9 and miR-181A of breast cancer-derived exosomes activate the JAK/STAT signaling pathway by targeting SOCS3 and PIAS3, respectively, and promote the amplification of eMDSCs, thereby significantly inhibiting T cell proliferation, stimulating T cell apoptosis, and accelerating tumor growth in vivo and in vitro [43].

Programmed death-1 (PD-1) is one of the co-inhibitory immune checkpoint (ICP) receptors induced by T cell activation. It is widely expressed in tumor-infiltrating lymphocytes (TILs) (70.3%) [44] and interacts with programmed death-ligand 1 (PD-L1, CD274) to induce negative signal transduction of effector T cell activity. PD-1 acts as a mediator for tumor cells to survive by avoiding T cell killing [45,46]. High expression of PD-L1 is associated with poor prognosis for breast cancer [47,48]. Yufan Qiu, Yi Yang et al. found that PD-1 of activated T cell-derived exosomes could weaken THE TNBC immune evasion induced by PD-L1 and weaken anti-tumor immunity in TME [49,50].

The increased expressions of GRP78, PERK, ATF6, and IRE1α in tumor tissues suggest that ER stress is activated in breast cancer tissues. ER stress promotes the release of miR-27a-3p enriched exosomes in breast cancer cells and induces the up-regulation of PD-L1 expression in macrophages. It also promotes immune escape of breast cancer cells by activating the PTEN-AKT/PI3K pathway [51].

#### 3.1.2. Exosomes Promote Invasion and Metastasis of Breast Cancer Cells

Cancer-associated fibroblasts (CAF) are populations of fibroblasts found in primary and metastatic cancers, which are associated with the onset, progression and metastasis of tumors [52]. In the tumor microenvironment, CAFs-derived exosomes (CAFs-Exos) can drive tumor development and metastasis [53,54]. BC-derived exosomes miR-146a activates the Wnt/β-catenin pathway by negatively inhibiting TXNIP and promoting the transformation of normal fibroblasts (NFs) to CAFs [55]. CAFs-Exos can down-regulate the target gene LATS2 of miR-92, resulting in markedly increased miR-92 expression in breast cancer cells (BCCs) and subsequently enhanced nuclear translocation of YAP1. As YAP1 binds to the enhancer region of PD-L1, it enhances the transcriptional activity of PD-L1, induces apoptosis of T cells and damages the function of NK cells [56]. Promotes BC growth by suppressing immune function. In addition, miR-18b of CAFs-Exos induces epithelial-mesenchymal transition (EMT) of BC and promotes cancer invasion and metastasis by targeting TCEAL7 to activate the NF-κB pathway and promote nuclear Snail ectopic [57].

Hypoxia-induced independent ATM activation of DNA double-strand breaks (DSB) regulates autophagosomes (AP) accumulation by phosphorylation of BNIP3, and promotes exosome release by phosphorylation of ATP6V1G1 inducing lysosomal dysfunction, leading to the fusion of AP and Multi-vesicular body (MVB) in hypoxic CAFs. In addition, autophagy associated GPR64 rich in releasing exosomes stimulates the activation of noncanonical NF-κB signaling in tumor cells and contributes to malignant invasion and metastasis of breast cancer [58].

The miR-500A-5P of CAFS-Exos in BC patients can be transferred from adjacent CAFs to BCCs, and by binding to ubiquitin specific peptidase 28 (USP28), the miR-500A-5P of CAFs-Exos can considerably promote the proliferation and metastasis of breast cancer cells by giving it an aggressive phenotype [59].

Extra cellular matrix (ECM) interacts with cells to regulate proliferation, migration and differentiation. More and more studies have found that ECM also plays a positive role in tumor progression [60,61,62].

In BC, with excessive collagen deposition, cross-linking, and remodeling, tumor development and progression to metastasis are normally accompanied by an increase in internal and peripheral ECM hardness, which then affects a series of cellular responses, ultimately enhancing in vitro invasions and in vivo metastases through Integrin integrin and PI3K signaling [63], and increasing cell contractility [64]. The hardness of ECM can be effectively regulated by exosomal thrombospondin (THBS1), while exosomal THBS1 promotes FAK-mediated focal adhesion by binding FAK and matrix metalloproteinase-9 (MMP-9) to regulate the process dynamics. To improve the sclerosis-induced migration and invasion of ECM in breast cancer [65].

Under the influence of chemotherapeutic drugs, some tumor cells have developed additional abilities in addition to drug resistance, such as a greater ability to invade. The tyrosine kinase EphA2 belongs to the Eph receptor family and is highly expressed in tumor tissues compared to most normal adults [66]. Zicong Gao et al. found that drug-resistant cancer cells promote the invasion of sensitive breast cancer cells by secreting epha2-rich exosomes through Ephrin A1-dependent reverse ligand pathway [67].

Kangdi Li et al. found that survivin in BCCs-derived Exos can transform fibroblasts into myofibroblasts (CAF-like) by upregulating SOD1. In addition, SOD1 up-regulated fibroblasts could promote the proliferation of BCCs, EMT and stemness [68].

X-inactive specific transcript (XIST), a lncRNA, is significantly down-regulated in brain metastases from BC patients, and its expression level is inversely correlated with brain metastases. When XIST is lost, the secretion of exosome-miR-503 increases, triggering the M1-M2 phenotype transformation of microglia, enhancing the expression of immunosuppressive cytokines in microglia, and thereby inhibiting T cell proliferation, thereby promoting breast cancer brain metastasis [69].

#### 3.1.3. Exosomes Are Beneficial to the Metastasis of Breast Cancer by Promoting the Formation of Pre-metastasis Niche

Low expression of multiple let-7 gene family members has been found to be associated with human cancer and cancer stem cells, and a large number of studies have shown that let-7 gene promotes terminal differentiation and inhibits cancer during development [70]. Lin28B induces primary tumors to produce more acetaldehyde dehydrogenase (ALDH) breast cancer stem cells (BCSCs), which release tumor exosomes containing low let-7s, the latter causes Lin28B to induce neutrophil infiltration and N2 transformation to establish the pre-metastatic niche (PMN). Immunosuppression in the metastatic niche of the lung promotes breast cancer metastasis. Thus, high Lin28B and low let-7s in tumors may indicate poor prognosis and lung metastasis in breast cancer patients [71].

Estrogen receptor-positive breast cancer (ER^+^ BC) has a higher incidence of bone metastasis [72]. Kerui Wu et al. found that regulating mir-19A expression in breast cancer cells did not change its growth or migration ability in vitro, but after being wrapped and released by Exos, integrin-binding Sialoprotein (IBSP) can attract osteoclasts and aggregate in the bone, becoming an osteoclast rich environment, and assist exosomal miR-19a to be delivered to osteoclasts to induce osteoclast formation, promoting The formation of PMN, enhancing bone metastasis in ER^+^ BC [73].

By regulating PDCD4 protein level, exosome-miR-21 derived from BCCs can promote the internalization of osteoclasts, enhance differentiation and function, reduce bone mineral density, and contribute to the formation of PMN, thus accelerating the reconstruction of bone lesion microenvironment for bone metastasis [74].

#### 3.1.4. Exosomes Can Change its Metabolic Microenvironment to Promote the Development, Invasion and Metastasis of Breast Cancer

Cancer-associated stromal cells rely on glycolysis to deliver energy metabolites to cancer cells through monocarboxylate transporters (MCTs) during disease progression [75]. In contrast to normally differentiated cells, which rely primarily on mitochondrial oxidative phosphorylation to generate the energy needed for cellular processes, most cancer cells rely on aerobic glycolysis, a phenomenon known as the Warburg effect [76]. The reprogramming of energy metabolism to drive rapid cell growth and proliferation is a novel feature of cancer [77]. Miranda Y Fong et al. found that cancer cells secreted exosome miR-122 and down-regulated glycolytic pyruvate kinase (PK) to inhibit glucose uptake by niche cells in vitro and in vivo and promote tumor metastasis in distant organs (including brain and lung) [78].

Aspartate β-hydroxylase (ASPH) is highly expressed in more aggressive poorly differentiated tumors, but low in less aggressive moderately differentiated tumors. Qiushi Lin et al. found that ASPH promotes BCCs synthesis/release of Exos carrying proteins such as MMPs by activating Notch signaling in BC patients. Lead to its spread and metastatic growth [79]. Mir-105-mediated stromal cell metabolic reprogramming promotes sustained tumor growth by regulating the shared metabolic environment. Exosomal miR-105 secreted by BCCs induced by the MYC gene of cancer cells induces metabolic programming by activating MYC signaling in CAFs. CAFs can show different metabolic characteristics in response to changes in metabolic environment. When nutrition is sufficient, miR-105-reprogrammed CAFs enhance glucose and glutamine metabolism to provide energy to adjacent cancer cells. When nutrients are deprived and metabolic byproducts accumulate, these CAFs detoxify by converting metabolic wastes, including lactic acid and ammonium, into energy-rich metabolites [80].

## 4. Breast Cancer Drug Resistance and Exosomes

Trastuzumab combined with paclitaxel or docetaxel has been the standard first-line treatment for HER2-positive metastatic breast cancer (HER2^+^ BC) [81]. Clinical trials have shown that trastuzumab, a recombinant monoclonal antibody that targets the HER2 receptor, can significantly improve the overall survival and disease-free survival of women with early HER2^+^ BC [82]. However, due to the emergence of drug resistance, Approximately 65% of patient with HER-2^+^ BC did not respond to trastuzumab initial therapy, and approximately 70% of patients who initially responded experienced disease progression one year after initial therapy. This means that trastuzumab is affected by two types of therapeutic resistance: primary or inherent resistance; Secondary or acquired drug resistance [83], in which exosomes play an important role.

AFAP1-AS1 (lncRNA) is transferred to breast cancer cells after being encapsulated by exosomes, leading to up-regulation of HER-2 expression and induction of trastuzumab resistance in BCCs by promoting AUF1-mediated ERBB_2_ translation activation [84].

Pasquale Sansone et al. identified complete Mitochondrial genomes in circulating exosomes from patients with hormone therapy resistance (HTR) metastatic BC, which transfer mitochondrial DNA (mtDNA) to metabolically impaired cells to restore their metabolic activity. mtDNA exosomal transfer can act on tumor stem cell-like cells, mediating an increase in their self-renewal potential, leading to resistance to hormone therapy [85].

Qianxi Yang et al. found an acquired chemical resistance mechanism (Figure 2) [86]: chemotherapy activates the EZH2 / STAT3 axis in BCCs and then secretes exosomes rich in miR-378A-3p and miR-378D triggered by chemotherapy. These exosomes are taken up by BC cells surviving chemotherapy and activate WNT and NOTCH stem cell pathways by targeting DKK3 and NUMB, subsequently leading to the emergence of drug resistance.

## 5. Application of Exosomes as Biomarkers in Breast Cancer

So far, a cancer diagnosis is still the most commonly used method is based on biopsy, this means that the need to extract tumor histologic analysis for further [87], this invasive method not only has some damage on the patients and time-consuming may delay treatment, have potential safety risks in certain patients, is not suitable for monitoring the process of tumor. In addition, biopsies can increase the likelihood of metastasis, and some tumors can not always be biopsied [88]. Therefore, collecting patient body fluids for liquid biopsies has become a new cancer diagnosis method that has attracted a lot of attention.

At present, the main biological components of liquid biopsy are mainly circulating tumor cells (CTCs), circulating tumor DNA (ctDNA), the tumor-derived fraction of cell-free DNA (cfDNA) in plasma, RNA (mRNA, lncRNA, miRNA), extracellular vesicles, tumor "educated" platelets, proteins and metabolites [89]. As a subset of EVs, exosomes have been widely selected by researchers.

Recent approaches based on the real-time exosome tracking system suggest that Exos can act as effective vector-mediated transfer factors both in vitro and in vivo [90,91]. Most of the early studies on EVs isolation were carried out by obtaining materials from cell culture medium, and humoral derived EVs (such as urine, blood, malignant ascites) have great scientific and clinical significance. A large number of studies have proved that exosome surface proteins, endogenous proteins and nucleic acids isolated from body fluids of various cancer patients are expected to be biomarkers for prediction, diagnosis, staging, radiotherapy and chemotherapy treatment response and prognosis [92], such as glioma [93]; Esophageal cancer [94]; Gastric cancer [95]; Pancreatic cancer [96,97]; Hepatocellular carcinoma [98]; Colorectal cancer [99]; Lung cancer [100]; Lung adenocarcinoma [101]. Non-small cell lung cancer [102]; Bladder cancer [103]; Prostate cancer [104]; Ovarian cancer [105,106]; Breast cancer [107], etc. Table 1 shows exosomes as biomarkers of breast cancer.

## 6. Exosomes Are Used in the Treatment of Breast Cancer

### 6.1. Exosomes for Vaccine Production

Cancer cells can overexpress anti-phagocytic surface proteins to avoid the clearance of macrophages, which is called “Don’t eat me” signal [148]. Consequently, enhancing tumor immunity is expected to become the focus of tumor vaccine.

Lanxiang Huang et al. loaded human neutrophil elastase (ELANE) and Hiltonol (TLR3 agonist) as inducers of immunogenic cell death (ICD) into exosomes derived from α -lactalbumin (α-LA) engineered BC (Figure 3). An in situ dendritic cell (DC) vaccine, HELA-Exos, has been developed to enhance anti-tumor immunity in breast cancer. Hela-Exos can selectively induce ICD in breast cancer cells, thus contributing to in-situ infiltration and maturation of type one conventional DCs (cDC1s) and activation of CD8^+^ T cells in vitro [149], and improving tumor immunity.

Hypoxia usually occurs in the TME associated with breast cancer [150]. Oxygenation of tumor cells interferes with their proteome and genomic transformation, resulting in death [151], but oxygenation merely transforms the immunosuppressive disease into an immunologically permitted microenvironment and reduces extracellular adenosine-mediated tumor protection [152,153]. Studies have found that when promoting Oxygenated Water for the treatment of BC, the use of Oxygenated Water while increasing the use of tumor-derived Exos can significantly improve the anti-tumor immune response [154].

In more than 90% of human tumor tissues, Fibroblast activation protein (FAP)—α is overexpressed in CAFs, including colorectal cancer, melanoma, generalized muscle cancer, lung cancer, ovarian cancer, and breast cancer [155]. Therefore, FAP+CAF is an ideal interstitial target for immunotherapy of solid tumors. Hu et al., Shichuan Hu et al., prepared a tumor cell derived exosome-like nanovesicle expressing FAP (eNVs FAP). It inhibits tumor growth by inducing a strong and specific cytotoxic T lymphocyte (CTL) immune response against tumor cells and FAP+CAFs and reprogramming immunosuppressant TME in colon cancer, melanoma, lung cancer, and breast cancer models. In addition, eNVs FAP vaccine can promote tumor iron death by releasing interferon γ (IFN-γ) from CTL and consuming FAP+CAF, thus achieving the killing effect on tumor parenma and tumor stromal cells [156]. Due to its easy preparation, eNVs FAP vaccine is expected to be a large-scale anti-tumor vaccine.

### 6.2. Drug Delivery of Exosomes in Breast Cancer

Mesenchymal stem cells (MSCs) derived Exos can induce functions associated with dormancy in breast cancer. For example, Exos secreted by cell cycle quiescence and chemotherapy-resistant mesenchymal stem cells instruct breast cancer cells to progressively dedifferentiate into dormancy in the perivascular region of bone marrow [157].

Piaopiao Wang et al. delivered the exosome M1-Exos released by M1-type macrophages into tumor tissues as a carrier to encapsulate Paclitaxel (PTX), which promoted the repolarization of M2-type macrophages to M1-type and increased the expression levels of casparase-3 and pro-inflammatory Th1 cytokines. Enhance the antitumor effect of chemotherapy drugs [158].

S100A4 is an important protein that promotes tumor progression and metastasis [159,160,161]. Cationic bovine serum albumin (CBSA) has been used as a gene carrier to package siRNA to form stable nanoparticles for targeted treatment of lung metastases [162]. Liuwan Zhao et al. have developed a biomimetic exosome coated cationic bovine serum albumin (CBSA) conjugated siS100A4 nanoparticles that target lung PMN, enhance siRNA accumulation and inhibit S100A4 expression in lung neutrophils via exosome mediated organic prokaryotic proteins. Significantly inhibited lung metastasis of breast cancer [163].

Sonodynamic therapy (SDT) is a non-invasive and accurate method to treat tumors by accumulating ultrasonic sensitizers in tumor cells and tumor neovascular endothelial cells, generating cytotoxic reactive oxygen species (ROS) through ultrasonic (US) irradiation, and playing a targeted killing role. However, it is a difficult problem to realize the targeted delivery of non-toxic ultrasound sensitizers for deep tumors. Folic acid (FA) is abundantly expressed in cancer cells [164,165]. Thuy Giang Nguyen Cao et al. Loaded the ultrasonic sensitizer indocyanine green (ICG) into FA modified exosomes (Figure 4) [166]. A single intravenous injection can play an excellent targeted enrichment, thus promoting the therapeutic effect of SDT. 

Autophagy, as a type II programmed cell death, plays a crucial role in cancer with autophagy-related (ATG) proteins [167,168], which can regulate autophagy through the regulation of miRNA on the expression of key ATG proteins. Mingli Han et al. found that exosomal coated Mir-567 can directly target the inhibition of cancer-associated ATG5, thereby inhibiting breast cancer autophagy and improving trastuzumab chemotherapy sensitivity [169].

### 6.3. Exosomes for Targeted Therapy of BREAST cancer

#### 6.3.1. Targeted Therapy for Triple Negative Breast Cancer

Exosomes can efficiently transfer internal substances to surrounding cells through rich adhesion proteins that interact with cell membranes [170,171]. After modifying it by various means, it can achieve non-toxic and high-accuracy targeted therapy.

Triple negative breast cancer is an invasive subtype of breast cancer that lacks expression of progesterone receptor (PR), estrogen receptor (ER) and human epidermal growth factor receptor (HER-2), accounting for 24% of newly diagnosed breast tumors [172]. Currently, EGFR-targeted therapy and hormone therapy do not respond to TNBC, and often develop resistance to chemotherapy [173]. Due to tumor heterogeneity and long-term lack of effective treatment other than chemotherapy, it is the subtype with the most adverse outcome [174]. EGFR was highly expressed in about 70–78% of basal TNBC samples [175]. Exos modified by GE11 peptide or EGF can be delivered to EGFR-expressing breast cancer cells by targeted delivery, and EGFR-expressing breast cancer cells can be targeted by systemic intravenous injection of exosomes coated with let-7a miRNA [176].

Cholesterol-modified RNA are asymmetric oligonucleotides with hydrophobic modifications, which can improve their stability and promote cellular internalization [177]. Chunai Gong et al. modified exosomes with deintegrin and metalloproteinase 15 (A15) to obtain targeted exosomes (A15 Exo), which were loaded with in vitro and in vivo therapeutic doses of doxorubicin (Dox) and cholesterol-modified miRNA 159 for combined targeted delivery to TNBC cells for treatment of TNBC.

To promote growth, cancer cells show an increased need for iron compared to normal cells. This iron dependence makes cancer cells more prone to iron-catalyzed necrosis, known as iron death, a modulatory form of iron-dependent cell death driven by excess lipid peroxidation, associated with the development and therapeutic response of various tumor types [178,179,180]. This may be the future direction of targeted therapies for refractory cancers [181].

Erastin is a low molecular weight chemotherapy agent that induces ferroptosis, but its use is limited by its poor water solubility and renal toxicity [182]. Mengyu Yu et al. prepared a targeted delivery nanobiologic drug for triple negative breast cancer by targeting exosomes derived from HFL-1 (human fetal lung fibroblasts) with FA modification and loading irastatin into the Exos [183]. The induction promoted ferroptosis of TNBC cells, accompanied by intracellular glutathione consumption and reactive oxygen species overproduction, achieving a strong anti-tumor effect.

#### 6.3.2. Targeted Therapy for HER2-Positive Breast Cancer

Anti-HER2 therapy significantly improves survival in patients with HER2-positive BC, but false-positive and false-negative laboratory test results are an important issue due to its high cost and potential toxicity [184]. Xiaojing Shi et al developed the Synthetic Multivalent Retargeted Exosome (SMART-Exo) by modifying exosomes with two different surface display monoclonal antibodies against human CD3 and human HER2. Has dual targeting of T cell CD3 and breast cancer-associated HER2 receptor. Efficient and specific HER2-positive BC therapy is achieved by redirecting cytotoxic T cells to attack breast cancer cells expressing HER2 [185].

The peripheral blood metastasis process of cancer cells includes: Neoangiogenesis, intravasation, circulation in the peripheral blood, extravasation, distant colonization and growth. are “seeds of dandelion”, so it is very important to target and eliminate CTCs.

Kaiyuan Wang et al., based on exosomes, co-encapsulated the conjugates of reactive oxygen species (ROS) -responsive sulfide linked paclitaxel-Linoleic acid conjugates (PTX-S-LA) and cucurbitin B (CuB) into polymer micelles. Exosome membrane (EM) was used to modify the nanoparticles. An exosome-like sequential bioactivating prodrug nanoplatform (EMPC) has been developed for the targeted treatment of breast cancer. They found that a variety of specific surface adhesion molecules, such as CD44, in the membranes of cancer-derived exosomes can mediate homotypic targeting by tracking and capturing CTCs through the adhesion molecule CD44. After CTCs uptake, CuB release can not only down-regulate the FAK/MMP signaling pathway to inhibit tumor metastasis, but also significantly increase intracellular ROS level and induce ROS responsive PTX-S-LA sequential bioactivation. In vitro and in vivo results showed that EMPCs not only exhibited amplifiable precursor drug bioactivation, prolongation of blood circulation, selective targeting of homotype tumor cells and enhancement of tumor penetration, but also regulated tumor metastasis inhibition through CTCs clearance and FAK/MMP signaling pathway (Figure 5) [186]. Selecting breast cancer-derived exosomes for the purpose of homologous targeting tracking may contain substances that promote the occurrence and development of tumors even if they have a good matching effect. Therefore, this risk cannot be avoided.

#### 6.3.3. Targeting Immune Cells

Macrophages play an important role in regulating tumor immune microenvironment. M1 macrophages can inhibit tumor growth, while M2 macrophages can promote tumor growth, invasion, migration and angiogenesis [187,188]. The phenotype of polarized M1-M2 macrophages can be reversed to some extent in vitro and in vivo [189].

Subir Biswas et al. found that exosomes specifically secreted by tumor-acclimated MSCs can promote the transformation of monocytic myeloid-derived suppressor cells (M-MDscs) into macrophages and acquire M2-like tumor-promoting properties [190]. MiR-138–5p of breast cancer-derived exosomes is delivered from breast cancer cells to tumor-associated macrophages, down-regulating lysine demethylase 6B (KDM6) histone demethylase transferase expression, inhibiting M1-type macrophage polarization and promoting M2-type polarization [191]. LncRNA BCRT1 is highly expressed in BC tissues and can competitively bind to miR-1303 to prevent the degradation of miR-1303 target gene PTBP3, which acts as a tumor promoter in BC. Meanwhile, lncRNA BCRT1 can further accelerate the progression of BC by promoting the M2 polarization of macrophages through exosome-mediated metastasis [129].

Therefore, inhibition of M2 macrophages and repolarization of M2 macrophages to M1 macrophages are common strategies for tumor treatment [192].

Sagar Rayamajhi et al. found that Exos derived from immune cells can simulate immune cells targeting cancer cells, so they used exosomes secreted by mouse macrophages to hybridise with synthetic liposomes, and designed vesicles with the size of exosomes and diameter less than 200 nm. Named Hybrid Exosome (HE), it is loaded with water-soluble doxorubicin to achieve targeted killing [193].

However, studies have proved that the half-life of immune cell-derived exosomes is short and the effect is limited. Therefore, it is possible to operate in engineering modification. This can be achieved through genetic modification and traditional chemical cross-linking, modifying exosomes with specific ligands, antibodies, or immunostimulating/inhibitory molecules [194].

M1-derived Exos have the ability to repolarize M2-derived macrophages into M1 [195]. It is well known that the tumor microenvironment is weakly acidic [196]. Weidong Nie et al. constructed a generalized exosomal nanobiological conjugate produced by ph-responsive macrophage M1 that could target tumor cells, at the same time on its surface binding with anti-tumor effect of antibodies (Figure 6) [197]. The nanobiotic binding of exosomes can play an anticancer effect through the phenotype of repolarized macrophages and synergistic antibody action.

## 7. Conclusions

Exosomes originate in various forms of intracellular endocytosis and can be isolated from conditioned cell culture media and biological fluids by a variety of methods. Similar to the composition of cell membrane derived from exosomes, it has similar and powerful function of biological information exchange between cells. Exosomes can carry or load a variety of substances inside, including nucleic acids, proteins, lipids, etc. In addition to participating in normal physiological functions, these substances also have an inseparable relationship with the occurrence, development and metastasis of breast tumors [198]. Exosomes derived from breast cancer can help tumor cells escape and weaken cellular immunity. Exosomes released by cancer-associated fibroblasts in the tumor microenvironment promote breast cancer cell invasion and metastasis. In the metastasis of breast cancer, exosomes can change the metabolic microenvironment and form premetastasis niche to enhance distant metastasis of cancer cells. Exosomes are also involved in the generation and transmission of drug resistance. Although exosomes promote the development of breast cancer, if it can be used correctly, it will have a positive effect on the treatment of breast cancer. Painless and injury-free biological predictors are one of the goals pursued by clinicians. Researchers have found multiple biomarkers for breast cancer screening, tumor development, drug resistance and prognosis in exosomes isolated from patient body fluids, which are of high clinical value [199]. The prevention of disease is often more important than the treatment after the occurrence of disease. It is very effective to reduce the risk of cancer by improving the immune response of the body against tumor by using exosomes. Exosomes are nanoscale in diameter, and due to their strong biocompatibility from cell membrane, they can easily pass through various physiological barriers, such as blood-brain barrier [200], enabling drug transfer. In the context of bioengineering and medical joint research, exosome and breast cancer research is in full progress [201]. Exosomes are modified by exogenous exosomes to obtain targets, so as to jointly realize precise drug transport, reduce systemic toxic reactions and enhance tumor tissue enrichment. The spread of cancer cells into the blood greatly increases the chances of distant metastasis. It is possible to capture and eliminate circulating tumor cells in the blood with the help of modified exosomes. M1-type macrophages have an inhibitory effect on tumors, while M2 is completely opposite. The intervention of exosomes can repolarize M2-type macrophages into M1-type and improve the progression of breast cancer. In summary, although related to breast cancer progression, exosomes are also a new platform for breast cancer treatment, showing high application potential. 

## Figures and Tables

**Figure 1 membranes-12-00775-f001:**
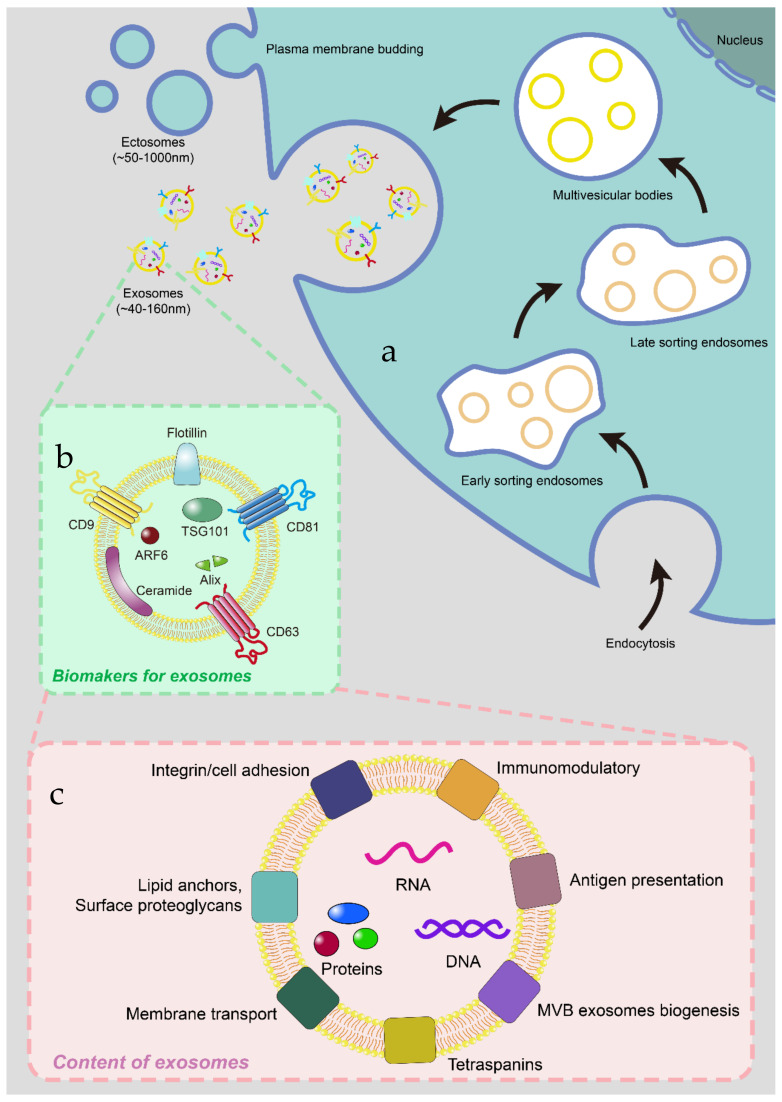
(**a**) Exosomes are extracellular vesicles between 40 nm and 160 nm in diameter that begin with endocytosis and then mature and exhaled in the cell. (**b**) Its biomarkers include CD9, CD63, CD81, flotillin, ceramide, ARF6, TSG101 and Alix. (**c**) Its contents include various nucleic acids, proteins and lipids.

**Figure 2 membranes-12-00775-f002:**
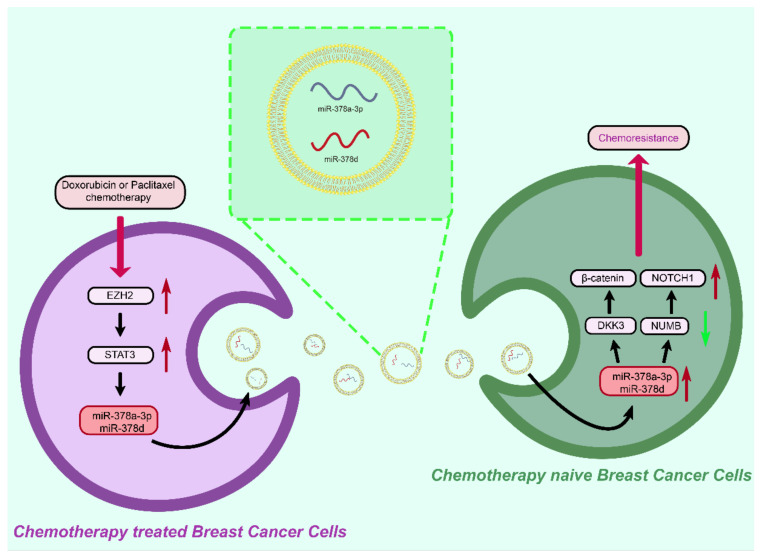
After activation of EZH2/STAT3 pathway, miR-378a-3p and miR-378d expression is increased in breast cancer cells treated with chemotherapy, which is released through exosome binding and endocytosis by surviving BC cells, resulting in chemoresistance.

**Figure 3 membranes-12-00775-f003:**
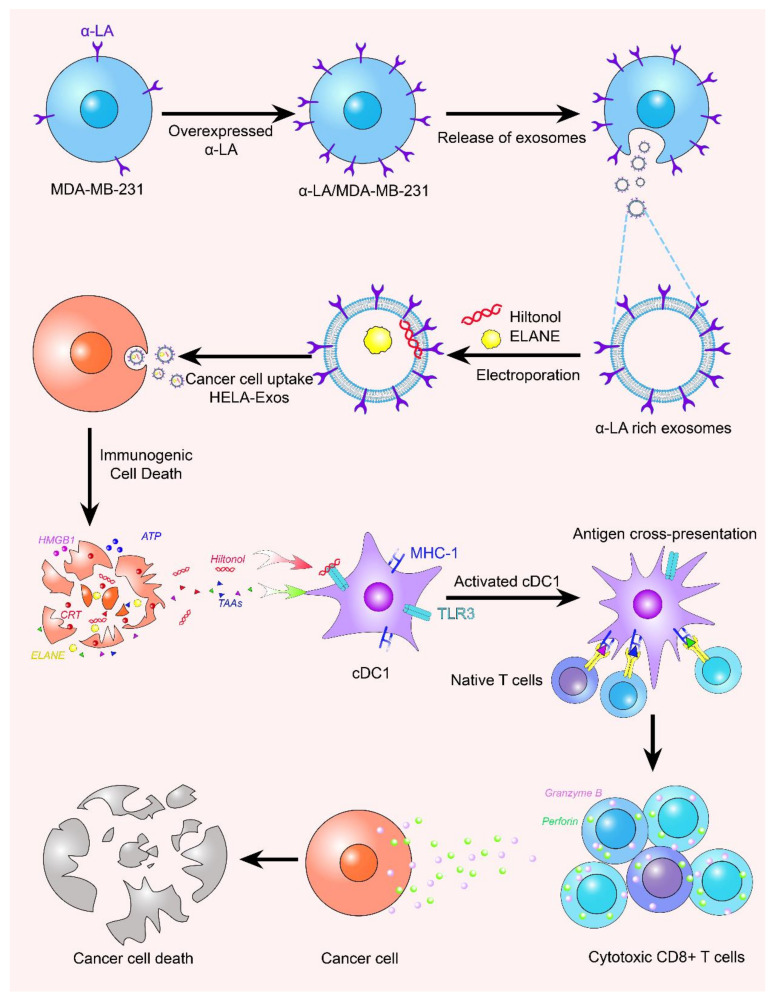
The expression of α-lactalbumin was low in breast cancer cell line MDA- MB-231. After overexpression treatment, the exosomes released by it had the characteristics of α-lactalbumin overexpression. Through electroporation, ELANE and hiltonol are mixed into the interior, and the uptake of cancer cells induces ICD. Subsequently, various substances released activate cDC1s, causing the activation of CD8^+^T cells, and then killing cancer cells.

**Figure 4 membranes-12-00775-f004:**
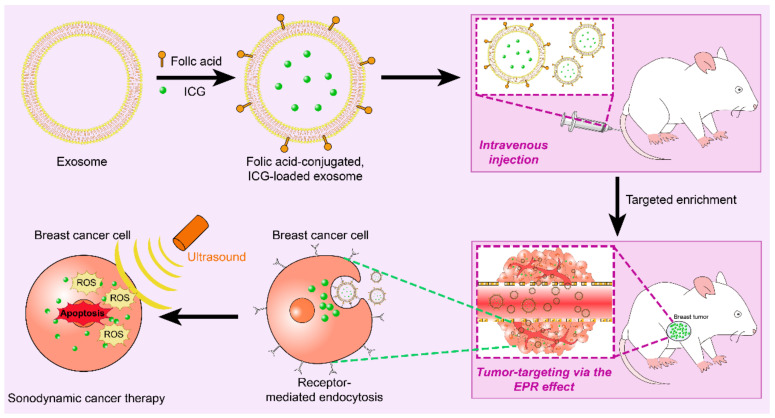
The exosomes were modified by FA and loaded with ultrasonic sensitizer ICG. After injection into the blood, they targeted the tumor tissue of mice and achieved local enrichment. Kill tumor tissue accurately.

**Figure 5 membranes-12-00775-f005:**
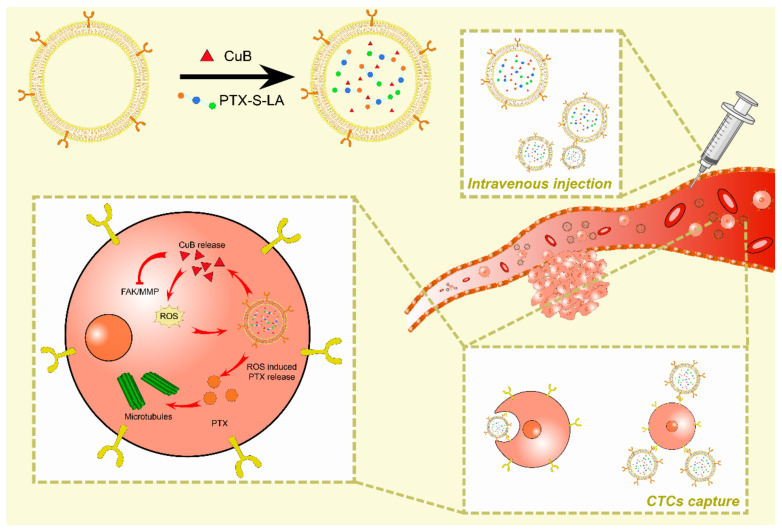
Exosomes from tumors have related proteins that can mediate homologous targeting on the surface. After loading CuB and PTX-S-LA into the exosomes. The exosomes can achieve targeted capture of homologous proteins with CTCs after intravenous injection into the blood. After exosome uptake by CTCs, CuB targeted FAK/MMP pathway in CTCs is released to inhibit tumor metastasis, and ROS is also released, which promotes sequential activation of PTX-S-LA and releases PTX to inhibit tumor growth.

**Figure 6 membranes-12-00775-f006:**
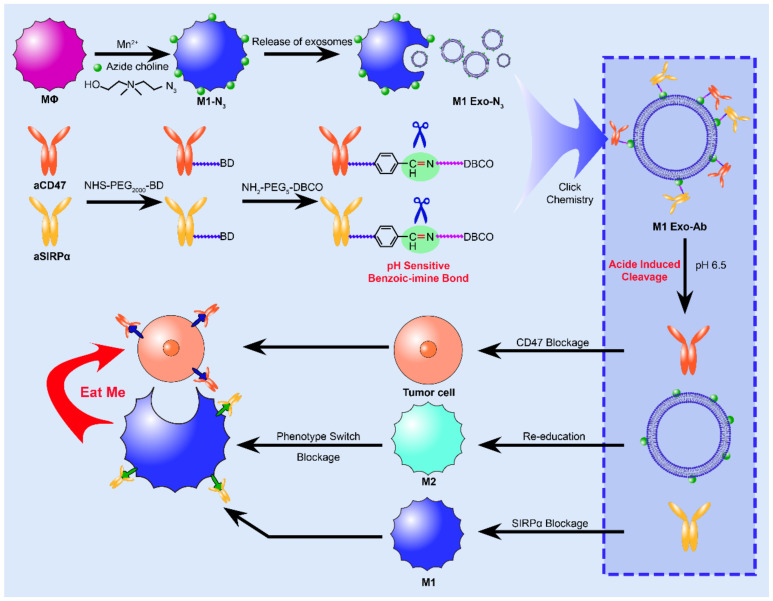
Mn^2+^ was used as an inducer to polarize macrophages into anti-tumor M1 macrophages. At the same time, the membrane is modified with azide groups through the inherent biosynthesis and metabolic incorporation of phospholipids. Then the azide modified M1 Exo was coupled with dibenzocyctocene (DBCO) modified anti-CD47 antibody (aCD47) and pH-sensitive anti-signal regulatory protein alpha (SIRPα) antibody (aSIRPα). After acid-induced division, antibodies are separated from exosomes, which repolarize M2-type macrophages into M1-type macrophages. ACD47 acts on CD47 overexpressed by tumor cells to resist the “don’t eat me” signal. Meanwhile, aSIRPa acts on SIRPA of macrophages to prevent phagocytic cells from losing their phagocytic ability. And then the phagocytes will eat the tumor cells.

**Table 1 membranes-12-00775-t001:** Breast cancer biopredictors. (The unlabeled biomarkers were all blood from breast cancer patients).

Biomarker	Clinical Significance
Positive correlation: miR-1246, miR-21 [108]; miR-202 [109]; has-miR-21–5p [110]; miR-106a-3p, miR-106a-5p, miR-20b-5p, miR-92a-2-5p (in plasma); miR-106a-5p, miR-19b-3p, miR-20b-5p, miR-92a-3p (in serum) [111]; let-7b-5p, miR-122-5p, miR-146b-5p, miR-210-3p, miR-215-5p [112]; GPC-1, ADAM10, GLUT-1 (CM of BCCs) [113]; RALGAPA2, PKG1, TJP2 [114]; miR-424, miR-423, miR-660, let7-I (Urine of BC patients) [115]. Negative correlation: miR-21 [116]	The correlation with breast tumorigenesis can be used for tumor screening.
miR-1976 [117]; CD82 [118]; miR-363-5p [119].	Negative correlation with breast tumorigenesis.
miR-1910-3p [120]; circHIF1A [121]; circ_0001142 (CM of BCCs) [122] (It also promotes polarization of macrophages); Hsa-miR-576-3p(brain metastases) [123]; miR-200c, miR-141 [124]; CD105 [125]; miR-370-3p [126]; miR-7641 [127].	Positive correlation with breast tumor metastasis.
Negative correlation: miR421, miR128-1, miR128-2 [128]; lncRNA BCRT1 (CM of BCCs) [129]; lncRNA AFAP1-AS1 [84]; lncRNA SUMO1P3 [130]; NGF [131]; circPSMA1 (CM of TNBC) [132]; UCH-L1 (before chemotherapy) [133]; miR-21, miR-200c (Tears from a patient with metastatic BC) [134]; lncRNA AC073352.1 (CM of BC) [135]; lncRNA H19 [136]; XIST [137]; has_circ_0000615 [138]; miR-223-3p [139]; miR-1910-3p [120]. Positive correlation: miR-148a [140]; miR-188-5p [141]	Prognostic correlation of breast cancer.
Del-1 [142]	The increased expression level after breast cancer operation suggests early recurrence.
miR-770 (CM of TNBC) [143]	High expression inhibits migration and invasion, and inhibits doxorubicin resistance in TNBC cell lines.
LncRNA HOTAIR [144]	Positive correlation with HER2 expression in tumor tissue.
GSTP1(CM of drug-resistant BCCs) [145]	High expression indicates transfer resistance.
TRPC5 [146]	The increased expression after chemotherapy suggested an increase in acquired chemical resistance.
LDH-C4 [147]	It was negatively correlated with drug therapy and positively correlated with relapse of BC.

## Data Availability

The datasets generated and/or analyzed during the current study are available from the corresponding author upon reasonable request.

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
