# Peer review of "Exosomes: Small Vesicles with Important Roles in the Development, Metastasis and Treatment of Breast Cancer"

_membranes, 2022, doi:10.3390/membranes12080775_

Round 1

Reviewer 1 Report

1. In section 3 "Exosomes and breast cancer microenvironment", it was well described that tumor cell derived exosomes are beneficial to tumor occurrence, development and metastasis. However, in section 6.3.2. "Targeted therapy for HER2-positive breast cancer", exosomes from tumors were used in a study of targeted therapy for this type of breast cancer. How do you evaluate the use of tumor cell-derived exosomes for cancer treatment ignoring the benefits of these nanoparticles to the tumors?

2. Figure 4 was based on the study using a mouse model. So, the figure should include cartoon mice other than human being.

3. The "Conclusion" part should be based on and close to the main body of text. Rewrite this part is recommended. 

4. Simplifying and regrouping table 1 is recommended. 

Author Response

Point 1: Simplifying and regrouping table 1 is recommended.

Response 1: The author has reorganized Table 1 for clinical significance.

Point 2: Figure 4 was based on the study using a mouse model. So, the figure should include cartoon mice other than human being.

Response 2: The author has redrawn Figure 4.

Point 3: In section 3 "Exosomes and breast cancer microenvironment", it was well described that tumor cell derived exo-somes are beneficial to tumor occurrence, development and metastasis. However, in section 6.3.2. "Targeted therapy for HER2-positive breast cancer", exosomes from tumors were used in a study of targeted therapy for this type of breast cancer. How do you evaluate the use of tumor cell-derived exosomes for cancer treatment ignoring the ben-efits of these nanoparticles to the tumors?

Respose 3: The author has added a new commentary to this section.

Point 4: The "Conclusion" part should be based on and close to the main body of text. Rewrite this part is recommended.

Respose 4: The author has rewritten the Conclusion.

Reviewer 2 Report

The authors provided a well-documented overview of Ev's role and application in Breast cancer. In addition to this, the review could represent a really interesting point of view in a field so dynamic and rich in potential future applications. The field of research focused on exosomes is in continuous evolution and even if the article is well written, the introduction section could be improved with a more general point of view about the application of EVs research in other fields of research adding some recent works related to the importance of exosomes in other diseases (PMID: 32932746). 

I hope that my comments could be useful and I look forward to reading the revised version of the paper.

Good luck.

Author Response

The authors provided a well-documented overview of Ev's role and application in Breast cancer. In addition to this, the review could represent a really interesting point of view in a field so dynamic and rich in potential future applications. The field of research focused on exosomes is in continuous evolution and even if the article is well written, the introduction section could be improved with a more general point of view about the application of EVs research in other fields of research adding some recent works related to the importance of exosomes in other diseases (PMID: 32932746). 

The authors have added the importance of exosomes in other diseases in the introduction section.